# National snakebite project on capacity building of health system on prevention and management of snakebite envenoming including its complications in selected districts of Maharashtra and Odisha in India: A study protocol

Rahul K. Gajbhiye[1]*, Itta Krishna Chaaithanya[2], Hrishikesh Munshi[1], Ranjan Kumar Prusty[3], Amarendra Mahapatra[4], Subrata Kumar Palo[4], Sanghamitra Pati[5], Arun Yadav[6], Manohar Bansode[7], Shashikant Shambharkar[8], Kanna Madavi[9], Himmatrao S. Bawaskar[10], Smita D. Mahale[11]

1 Clinical Research Laboratory, ICMR-National Institute for Research in Reproductive and Child Health (NIRRCH), Mumbai, Maharashtra, India, 2 Molecular Immunology and Microbiology Laboratory, ICMR-National Institute for Research in Reproductive and Child Health (NIRRCH), Mumbai, Maharashtra, India, 3 Department of Biostatistics, ICMR-National Institute for Research in Reproductive and Child Health (NIRRCH), Mumbai, Maharashtra, India, 4 Epidemiology Division, ICMR- Regional Medical Research Centre (RMRC), Bhubaneswar, Odisha, India, 5 ICMR- Regional Medical Research Centre (RMRC), Bhubaneswar, Odisha, India, 6 Public Health Department Maharashtra, Government of Maharashtra, Mumbai, Maharashtra, India, 7 Sub District Hospital Shahapur, Government of Maharashtra, District Thane, Maharashtra, India, 8 Nutrition Bureau, Deputy Director Health Services Office, Government of Maharashtra, Nagpur, Maharashtra, India, 9 Sub District Hospital Aheri, Government of Maharashtra, District Gadchiroli, Maharashtra, India, 10 Bawaskar Hospital and Research Center, Mahad, District Raigad, Maharashtra, India, 11 ICMR-National Institute for Research in Reproductive and Child Health (NIRRCH), Mumbai, Maharashtra, India

* gajbhiyer@nirrh.res.in

## Abstract

### Background

Snakebite envenoming (SBE) is an acute, life-threatening emergency in tropical and sub-tropical countries. It is an occupational hazard and a major socioeconomic determinant. Limited awareness, superstitions, lack of trained health providers, poor utilization of anti-venom results in high mortality and morbidity. India is the snakebite capital of the world. Yet, information on awareness, knowledge, and perceptions about snakebite is limited. Data on capacity building of health systems and its potential impact is lacking. Recommended by the National Task Force on snakebite research in India, this protocol describes the National Snakebite Project aiming for capacity building of health systems on prevention and management of snakebite envenomation in Maharashtra and Odisha states.

### Methods

A cross-sectional, multi-centric study will be carried out in Shahapur, Aheri blocks of Maharashtra, and Khordha, Kasipur blocks of Odisha. The study has five phases: Phase I

**Data Availability Statement:** No datasets were generated or analysed during the current study. All relevant data from this study will be made available upon study completion.

**Funding:** This project is funded by the Indian Council of Medical Research (ICMR), (no: 58/6/NTF-Snakebite/2019-NCD-II). The funders had no role in study design, data collection and analysis, decision to publish, or preparation of the manuscript.

**Competing interests:** The authors have declared that no competing interests exist.

**Abbreviations:** ANM, Auxiliary Nurse Midwife; ASHA, Accredited Social Health Activist; CHC, Community Health Center; CPR, Cardiopulmonary Resuscitation; CTRI, Clinical Trials Registry of India; DALY, Disability Adjusted Life Year; DH, District Hospital; FGD, Focus Group Discussion; HCW, Healthcare Worker; ICMR, Indian Council of Medical Research; ICVI, Item Level Content Validity Index; IEC, Information Education Communication; INSP, ICMR National Snakebite Project; IPHS, Indian Public Health Standards; MO, Medical Officer; MPW, Multi Purpose Worker; MRHRU, Model Rural Health Research Unit; NIRRCH, National Institute for Research in Reproductive and Child Health; NTD, Neglected Tropical Diseases; NY, USA, New York, United States of America; PHC, Primary Health Center; RH, Rural Hospital; RMRCB, Regional Medical Research Center, Bhubaneshwar; SBE, Snakebite Envenoming; SC, Sub Center; SDH, Sub District Hospital; SEARO, South East Asia Regional Office; SMOG, Simple Measure of Gobbledygook; STG, Standard Treatment Guidelines; TAC, Technical Advisory Committee; WHA, World Health Assembly; WHO, World Health Organization.

involves the collection of retrospective baseline data of snakebites, facility surveys, and community focus group discussions (FGDs). Phase II involves developing and implementing educational intervention programs for the community. Phase III will assess the knowledge and practices of the healthcare providers on snakebite management followed by their training in Phase IV. Phase V will evaluate the impact of the interventions on the community and healthcare system through FGDs and comparison of prospective and baseline data.

## Discussion

The National Snakebite Project will use a multi-sectoral approach to reduce the burden of SBE. It intends to contribute to community empowerment and capacity building of the public healthcare system on the prevention and management of SBE. The results could be useful for upscaling to other Indian states, South Asia and other tropical countries. The findings of the study will provide critical regional inputs for the revision of the National Snakebite Treatment protocol.

## Trial registration

Registered under the Clinical Trials Registry India no. CTRI/2021/11/038137.

## Introduction

Snakebite envenoming (SBE) is one of the neglected tropical diseases (NTD) leading to around 81,410 to 137,880 deaths from 1.8 million to 2.7 million cases globally [1, 2]. SBE affects around 400,000 people every year causing permanent physical or psychological disabilities including blindness, amputation, and post-traumatic stress disorder [3]. It is estimated that in countries with a frail health system and scarcity of anti-venom, one death occurs every five minutes and four more people are disabled permanently due to SBE [4]. World Health Organization (WHO) classified SBE as a high-priority neglected tropical disease (NTD) in 2017, and subsequently, in May 2018, Seventy-first World Health Assembly (WHA) adopted a resolution providing a strong mandate to WHO for global actions on reducing the burden of SBE [5]. On May 23, 2019, WHA launched its roadmap to reduce the death and disability from snakebite by 50% by 2030 [4]. The strategy focuses on prevention of snakebite; provision of safe and effective treatment; strengthening health systems; and increased partnerships, coordination, and resources. Community education for seeking early and appropriate treatment, accelerating development and stockpiles of anti-venom, and stabilizing the market for snakebite treatments are also recognized as important aspects of this strategy [4, 6].

The recent national mortality survey [7] estimated that India had 1.2 million snakebite deaths (average 58,000/year) from 2000 to 2019 which is an increase of about 8000 cases/year compared to the earlier estimated survey (2001–2003). However, only a 10% coverage of the actual snakebite burden being captured in the government data in Maharashtra was also reported indicating gross underestimation of morbidity and mortality in Maharashtra [8]. The majority of the deaths occurred at home in the rural areas with half of the deaths happening between 30–69 years of age. Eight states (Madhya Pradesh, Odisha, Uttar Pradesh, Bihar, Jharkhand, Rajasthan, Gujarat, and Andhra Pradesh including Telangana) of India shared the burden of about 70% snakebite deaths from 2001 to 2014 [8]. A nationally representative mortality survey conducted by Mohapatra B et al 2011, included Maharashtra in the high prevalence of snakebite envenomation (SBE) group with an age standardized mortality rate of 3.0 per 100000 people. Except the high prevalence states, for the rest of the country the age

standardized mortality rate was 1.8 per 100000 people during the same period. In the same study, Maharashtra was also reported to have the fifth largest number of deaths due to snakebite among the high prevalence states [9]. Snakebites usually occur in geographically remote areas and hence their burden either goes unnoticed or is underreported [10]. From 2003 to 2015, the million death study estimated about 154,000 snakebite deaths in both private and public hospitals. However, the government reported only 15,500 hospitals deaths indicating that only about 10% of the expected deaths were captured [7].

SBE is labelled as a disease of poverty [11]. Agricultural and migrant workers, tribes, hunters and often, the earning members of the family are the victims. So it is not only a public health problem but also a major socio-economic determinant in India [12]. A study conducted in the tribal region of Maharashtra, India, reported 4.5% case fatality rate [13] while a study in rural Nepal reported an annual incidence of 1162/100000 with an annual mortality rate of 162/100000 due to SBE [14]. A meta-analysis of data from 41 Sub-Saharan African countries estimated the annual burden of SBE related deaths as 1.03 million DALYs, higher than the burden of many NTDs globally [15]. Similarly, a study in West Africa revealed a higher burden of SBE compared to other NTDs found in 16 countries and labelled SBE as underappreciated [16].

SBE burden coupled with a lack of awareness amongst the Medical Officers (MOs) and other healthcare providers about the National Snakebite Management Protocol (2009) and Standard Treatment Guidelines (STG, 2017) creates major hurdles in management of SBE [13]. An irrational usage of anti-venom skin test was also reported in the tribal block of Dahanu, Maharashtra [17]. Primary Health Centers in rural India face a multitude of problems including the acute shortage of trained human resources, unavailability of anti-venom and emergency ventilation services [18]. Issues with public health facilities and geographical inaccessibility supplemented by superstitions and cultural beliefs regarding snakes force SBE victims to seek care from faith healers [19, 20]. Studies in rural and tribal areas have revealed inadequate knowledge about venomous snake identification and faith healers were the first choice for about 38–68% of people for SBE management [17, 21–24]. Most of the traditional and herbal methods of first aid and management of snakebite have been found to cause more harm than benefit [25, 26]. Proper knowledge of snakes and snake-bite management was also found to be either diminutive or absent in the majority of participants as reported by a study from Haryana [27]. Fear, negativity, and unfamiliarity regarding prevention and ambivalent opinions on health-seeking are known drivers of people's perception of snakes and snakebites [28]. These findings strongly suggest the need for community awareness on the prevention and first aid of snakebite and empowerment of frontline healthcare workers and Medical Officers for effective management of SBE in public healthcare settings.

Previously, the authors have established a successful model for the prevention and management of SBE at Dahanu block of Palghar district through the Model Rural Health Research Unit (MRHRU) [13, 21]. The model involved community awareness, training of healthcare workers, snake handlers, and faith healers, availability of anti-venom and, implementation of the National Snakebite Treatment Protocol. Indian Council of Medical Research (ICMR)-National Task Force Expert Group for 'Research on Snakebite in India' recommended upscale of the Dahanu model to a Health System Research project using a similar multi-sectoral approach for attaining the 2030 goal set by WHO. Based on those recommendations, this protocol was developed and it describes the National Snakebite Project that aims for prevention and management of SBE in selected districts of Maharashtra and Odisha.

The objectives of the study are:

I. To increase the awareness and empower the community on prevention, first aid, and early transport of snakebite patients to the nearest health facility

II.  To evaluate the healthcare providers regarding their knowledge and practices during management of SBE and understand the anti-venom distribution and utilization at public health facilities

III.  To empower the health system for management of SBE through the implementation of Standard Treatment Guidelines (STG) of the Government of India

IV.  To study the impact of the interventions on reducing the SBE mortality and morbidity

## Materials and methods

### Study scheme

The interventional study has retrospective, prospective, cross-sectional, and qualitative components and will be conducted in a community as well as hospital settings (Fig 1). It is divided into five phases (Fig 2). Phase I involves collecting retrospective data of snakebites, facility surveys, and focus group discussions (FGDs) in the community. Phase II is to increase awareness and empower the community on prevention, first aid, and early transport of snakebite patients to the nearest health facility. Phase III will be to evaluate the basic knowledge on snakebite management and anti-venom utilization among healthcare providers. Phase IV is to empower the health system for management of SBE and optimal utilization of anti-venoms through training programs and Phase V will focus on the impact evaluation of the health system and community interventions using FGDs and comparison of pre-and post-intervention data.

### Study setting

From 2001 to 2014, Odisha and Maharashtra bore the burden of about 40300 and 56000 snakebite deaths with an age-standardized death rate of 6.7 and 3.5 / 100000 respectively [8]. Deaths

| STUDY PERIOD (months) | | | | | | | | | | |
|---|---|---|---|---|---|---|---|---|---|---|
| TIMEPOINT | -24 | 0 | 3 | 6 | 9 | 12 | 15 | 18 | 21 | 24 |
| **Activities** | | | | | | | | | | |
| Retrospective data | ●—————● | | | | | | | | | |
| Pre-Intervention Focus Group Discussions | | | | ✓ | | | | | | |
| Facility Check Survey | | | ✓ | | | | | | | |
| Interviews of Medical Officers | | | | ✓ | | | | | | |
| Prospective data | | | | | | ●————————————————● | | | | |
| Post-Intervention Focus Group Discussions | | | | | | | | ✓ | | |
| **Interventions** | | | | | | | | | | |
| IEC Campaign | | | | | ●————————● | | | | | |
| Community Talks | | | | | ●————————● | | | | | |
| Training of trainers | | | | | ●——● | | | | | |
| Healthcare Provider Training | | | | | | ●————● | | | | |
| Periodic Visits to Health Facilities | | | | | | | ✓ | ✓ | ✓ | ✓ |
| **Impact Assessment** | | | | | | | | | | |
| Case Fatality Rate | | | | | | | | | | ✓ |
| Snakebite incidence | | | | | | | | | | ✓ |
| Complications due to SBE | | | | | | | | | | ✓ |
| Notification of snakebite cases | | | | | | | | | | ✓ |
| Implementation of Standard Treatment Guideliens, 2017 | | | | | | | | ●————————● | | |

**Fig 1. Timeline of the study.**

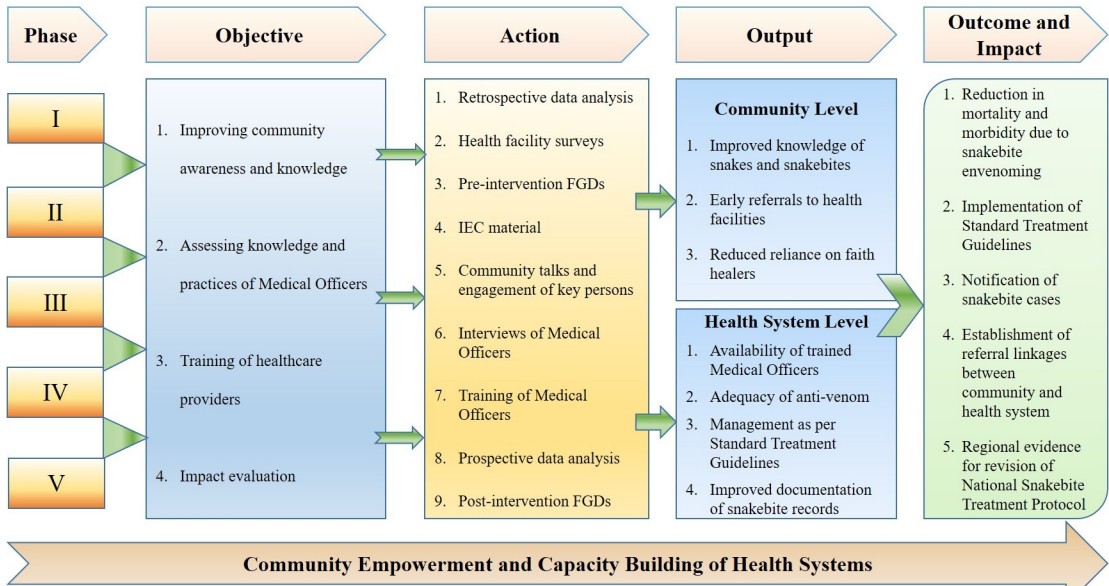

**Fig 2. The outline of the study.**

between the age of 5 to 14 years were greater in Odisha while deaths among females were prominent in Maharashtra [8]. SBE has been a chronic problem in these two states with an earlier county-wide survey (1941–1945) suggesting that compared to other states, snakebite mortality is a greater problem here as the majority of geographical areas are forested and infested with snakes [29]. The proposed study is a multi-center study that will be conducted in the West and East Zones of India. Under the West zone (Maharashtra state), two blocks namely, Shahapur in district Thane and Aheri in district Gadchiroli will be included. Under the East zone (Odisha state), two blocks namely, Khordha in district Khordha and Kasipur in district Rayagada will be included (Fig 3, Table 1). The population in the study areas lives with an

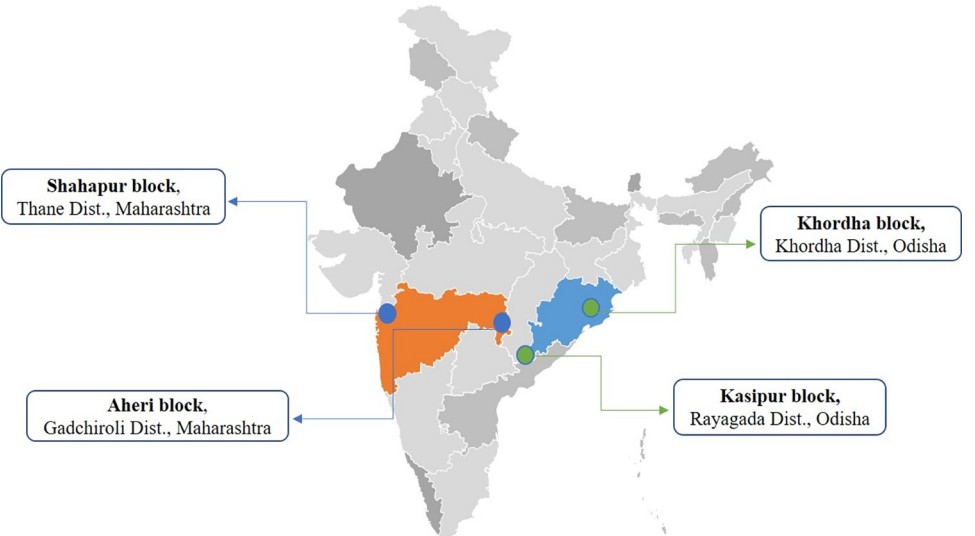

**Fig 3. Geographical location of study areas of Maharashtra and Odisha states in India.** Source: AMCHARTS SVG Maps; https://www.amcharts.com/svg-maps/?map=india2019. Permission obtained for use.

**Table 1. Demographic details of the study areas in Maharashtra and Odisha.**

| Parameter | Shahapur block | Aheri block | Khordha block | Kasipur block |
|---|---|---|---|---|
| Coordinates | 19.45˚N 73.33˚E | 19.41˚N 80.00˚E | 20.16˚N 85.66˚E | 19.22˚N 83.08˚E |
| Population | 314103 | 116992 | 139978 | 70542 |
| Living in rural areas | 77.5% | 87.5% | 67% | 100% |
| Scheduled Tribes population | 35.7% | 49.8% | 8.4% | 58.8% |
| Female literacy | 58.41% | 55.67% | 73.22% | 20.35% |
| Sex ratio | 957 | 986 | 935 | 1071 |
| Workers engaged in marginal activities* | 27.6% | 29.8% | 25.5% | 60% |

All data sourced from—Census of India 2011. Registrar General and Census Commissioner of India. https://www.censusindia.co.in/. Accessed 24 April 2022.

*Marginal activities provide livelihood for less than six months in a year. Proportions out of total working population in the block

absolute risk of ≥0.5% of dying due to snakebite before the age of 70 years with the risk being highest in Aheri and Kasipur [8].

## Framework for the study

**Roles and responsibilities.** The study coordinator will ensure the smooth and ethical conduct of the study. The site Principal Investigators and the project staff will be responsible for study site conduct. A Technical Advisory Committee (TAC) will be constituted steering the study. The TAC will comprise of national and regional experts in the area of snakebite management who have contributed to the development of STG, 2017. National and State Program Managers, Public Health Department members, social scientists, public health experts, herpetologists, forest departments will also be included in the study.

**Development of data collection tools and their validation.** The data collection tools will be developed by public health experts based on a review of available literature, standard treatment guidelines for snakebite, and consultation with the TAC.

*Retrospective and Prospective data form.* These two data collection forms will include questions about the basic demographic characteristics of the snakebite victim followed by details of the event of snakebite, signs and symptoms, first aid, and steps taken for the management including usage of anti-venom. The forms will have a separate 'Follow Up' section for transferred out patients. Hospital records will be used for form filling.

*Facility Survey Questionnaires.* Separate questionnaires will be developed for Primary Health Centre (PHC), Rural Hospital (RH) / Community Health Centers (CHC), and Sub District Hospital (SDH) / District Hospital (DH) to document staff positions, drugs, equipment, anti-venom availability and utilization, infrastructure, laboratory investigations, and Information Education Communication (IEC) materials and activities. The questionnaires will concur with the Indian Public Health Standards (IPHS) Guidelines and STG, 2017 for respective facilities.

*Focused Group Discussion (FGD) Guide.* The FGD guide will consist of open-ended, semi-structured questions to elicit the knowledge, awareness, and perceptions of people in the study area regarding snakes and snakebites, first aid, prevention methods, management, and health-seeking behavior. The FGD guide will be developed in English and later translated to local languages.

*Questionnaire for healthcare providers.* Separate questionnaires for peripheral workers and Medical Officers (MO) at facilities would be developed based on the STG, 2017 and WHO SEARO Guidelines for management of snakebites. They will try to capture the knowledge, perceptions, and experiences of healthcare providers regarding snakes and snake bites, anti-venom use, challenges encountered while managing snakebite cases at public health facilities.

All the developed tools will be validated through the following methods–

*Face Validity*. The TAC will critically review the tools to confirm that they have sufficient and pertinent information to gather the knowledge, awareness, and perceptions of the respondents. The panel will be asked to comment on the relevance, language, ease of understanding, and ability of the questions to elicit desired responses.

*Content validity*. Each item on the questionnaire will be rated based on a four-point Likert scale ranging from 'Not Relevant to 'Highly Relevant. Based on the rating given by experts, Item Level Content Validity Index (I-CVI) will be calculated. Items with an I-CVI value of 0.8 or more will be included in the final version [30].

*Construct validity*. A pilot study of the tools in a demographically similar population will be carried out to examine the degree to which the observations concur with the investigator's hypothesis and will make way for further revisions of the tools.

*Readability*. To confirm the understanding of the questions, the Simple Measure of Gobbledygook (SMOG) score will be used.

*Difficulty index*. Any question that is unclear to the participants will be identified and reworded by taking the percentage of correct responses overall. Questions unclear to 50% of participants will be rewritten.

In addition, all the research tools will be pre-tested in a similar population and revisions would be made based on the pre-test findings before actual use in the study.

**Phase I–Retrospective data collection, facility surveys, and FGDs in the community.** Two-year retrospective data (01 January 2020 to 31 December 2021) will be collected from all the public health facilities in study areas where snakebite management is expected to be provided. This includes all the Primary Health Centers, Rural Hospitals / Community Health Centers, Sub District Hospitals and District Hospitals. Medical Colleges where severe cases are referred from the study blocks will also be included to ensure quality retrospective data collection. Additional data on snakebite deaths will also be collected from local authorities, panchayat office, registrar of births and deaths so that maximum deaths are captured. A cross-sectional survey will assess the preparedness of the public health facilities for SBE management. Retrospective data and facility data will be collected and entered into an online application. The captured data will be verified at regular intervals by the study coordinator.

The qualitative component will involve conducting community FGDs in the study areas. Total 24 FGDs will be conducted, six per block (three male, three female). Each FGD will be restricted to 8 to 16 participants. Actual number of FGDs may vary according to the saturation of responses. FGDs will be conducted at common and acceptable community places. The Primary Health Centers (PHCs) in the block would be enlisted and one random village would be picked per PHC. The information about conduct of a FGD would be given to the villagers with the help of the frontline community healthcare workers and community leaders and 8–16 randomly selected participants would be invited to participate. Participants belonging to the community and above 18 years of age will be included for FGDs. Healthcare providers will be excluded.

**Phase II—Development and implementation of an educational intervention.**  Educational interventions in the form of printed IEC material will be developed based on the gaps identified during the FGDs and a review of the literature. The TAC will review the IEC material based on the relevance and ease of understanding for the target population. Commonly occurring words in the local language along with pictorial/cartoon messages will be used in the IEC material. Contact numbers of public health facilities will be included. Short information videos about immediate first aid and early referral will be developed and circulated among the community members. Videos regarding immediate care of a snakebite victim at the facility will be circulated among the healthcare providers. Pilot intervention procedures will be

implemented to know the community's understanding and accordingly further intervention procedures will be carried out by trained staff. The final revised version of the IEC will be assessed for ease of readability using the SMOG index. All educational tools will be validated with the help of the Technical Advisory Committee, herpetologists and subject experts.

IEC materials will be distributed at key places like village panchayats, forest departments, local faith healers, school teachers, community leaders, anganwadis, tribal residential schools, Sub-Centers (SCs), PHCs, and other health facilities. With the help of local health officials and ASHAs, community meetings and talks would be held at each study site during *gram sabha* (village gatherings), community program and religious events. Informative posters would also be put up at places where people gather routinely like tea stalls or local restaurants. Prevention measures to be employed for snakebite prevention in the communities will be as per the Standard Treatment Guidelines, (STG,2017).

**Phase III—Evaluation of the healthcare providers.** Assessment of the basic knowledge, awareness, and perceptions on snakebite management and preventions among healthcare providers will be conducted. All the Medical Officers and Peripheral Health Workers from the study blocks who agree to participate will be included in the study. Each block will have approximately 40 MOs; so nearly 160 MOs will be assessed during the study. Similarly, around 600 peripheral workers (150 per block) will also be evaluated. Actual numbers may vary as the MOs or healthcare workers are frequently transferred in the health system.

**Phase IV—Capacity building of healthcare providers.** The capacity building of healthcare providers will be done by periodic short term training programs. A team of national experts for snakebite management (Expert group STG, 2017) will provide training to master trainers at each study site. Total six master trainers, including two from State Public Health Department, two Senior MOs from SDH/DH, and two MOs from PHC/RH/CHC, will be trained per block. The training will involve lectures and practical demonstrations to cover all aspects of snakebite management including signs and symptoms of SBE, how to suspect/recognize snakebite, management of sever cases, anti-venom use, laboratory examinations, referral and discharge criteria. Knowledge gaps identified in phase III will also be covered in the training sessions.

All master trainers will be provided quick reference guides and treatment flyers, prepared as per the STG, 2017. The master trainers along with site investigators will conduct training programs for all the MOs every six months for two years. The national expert team will provide technical support for these regional trainings. Trained MOs will further train around 600 ASHAs, Auxiliary Nurse Midwives (ANMs) and Multipurpose Workers (MPWs) at their respective health facilities on first aid skills, immobilization techniques and cardiopulmonary resuscitation. Quick reference charts as per STG (2017) and training manuals in regional languages will be given to healthcare providers and also displayed in examination rooms and indoor wards of health facilities. Site investigators will make periodic visits to health facilities for supportive supervision, post-training follow up and grievance redressal.

**Phase V—Impact evaluation of interventions.** FGDs will be conducted in the study areas to evaluate the post-intervention community knowledge with the same methodology as discussed in Phase I, two months after the educational intervention. Pre- and post-training evaluations will be done to assess knowledge retention and the impact of training on the healthcare providers. Prospective data will be collected from the health facilities after the capacity building training of Medical Officers in study blocks for a period of one year. Incidence and case fatality rate in the study blocks will be calculated based on the prospective data. The impact of capacity building on the healthcare system will be assessed by comparing prospective data with the retrospective (pre-intervention) data.

## Ethics and dissemination

Ethics approval was obtained from the Institutional Ethics committee of ICMR- NIRRCH (Ref. No. D/ICEC/Sci-194/209/2021) and ICMR-RMRCB (ICMR-RMRCB/IHEC-2021/79). The study is registered under the Clinical Trial Registry India no. CTRI/2021/11/038137. Ethics committees of ICMR-NIRRCH and ICMR-RMRCB will supervise data monitoring and trial conduct. Written informed consent will be taken from each participant in the vernacular language before inclusion. Permissions are obtained from the Public Health Authorities for the implementation of the study. Approval of all investigators and Ethics Committees will be sought before any major modifications or amendments to the protocol are implemented. Minor changes will be approved by the investigators and the same would be notified to the Ethics Committees before implementation. The results from this study will be disseminated with local, state, and national health authorities, and the general population on the study website, social media as well as through scientific meetings, media reports and publications.

## Confidentiality and data storage

A central database to capture all the generated data will be developed and maintained at ICMR-NIRRCH, Mumbai. Qualitative data including audio recordings and transcripts will be kept separately and all the records that identify the participants will be kept confidential. Data will be secured as per the ICMR National Guidelines for Biomedical and Health Research Involving Human Participants [31]. Data entry and data verification will be carried out independently. Data will be backed up regularly in hard drives with sufficient memory space. RG will have ultimate authority over the study dataset.

## Statistical analysis

The baseline data will be analyzed separately for different strata (Gender-wise, Age-wise, seasonal variation, distribution of venomous and non-venomous snakebite, site of snakebites, and categorization of snakebite based on the sign and symptoms of envenomation). The differences across strata will be investigated using appropriate statistical tests based on the nature of the variables. The Chi square and Fisher's Test would be used to compare categorical variables while the independent t-test would be used to compare continuous variables. To check the factors associated with the incidence of snakebite, bivariate regression models will be used and results will be presented through odds ratios. The primary outcome will be reduction in the incidence of snakebite cases per 100000 population and a reduction in the case fatality rate in the study areas at the end of two years, measured by comparing the pre- and post-intervention data on the incidence of snakebite and mortality in the four study blocks. The secondary outcomes will include reduction in mean bite to needle time, reduction in the proportion of snakebite victims referred to higher facilities for management, increase in the proportion of MOs providing treatment as per STG 2017, mean increase in the number of vials of anti-venom administered to victims, reduction in the proportion of cases with severe complications including reduced kidney failure and amputation rate. Data from interviews of healthcare providers will be analyzed further for frequency. A summative index will be developed from the knowledge test of health workers. Two sample t-test will be used to compare the change in the knowledge of Medical Officers and the healthcare workers after the training. A p-value $< 0.05$ will be considered statistically significant. All the quantitative data will be entered and analyzed using the statistical package SPSS (version 26.0; IBM Corporation, Armonk, NY, USA).

Change in knowledge and awareness of the community will be assessed through pre-and post-intervention FGDs. Audios of the participants' discussion will be recorded and notes will be taken. Transcripts will be proofread and then translated to English by the project staff. The

transcriptions and translations will be cross checked by a senior project staff to ensure accuracy. A sample of the corrected translations will be read by the investigators to identify themes and subthemes using an inductive approach. Any additional themes or sub-themes after descriptive content analysis will also be identified. Disparities in the sub-themes or themes will be discussed by the investigators and consensus will be sought on the definitions of codes. Coding for the transcripts, based on the consensus definitions, will be carried out by two experienced staff members. All the coded transcripts will be reviewed for quality, consistency and accuracy. Broad themes based on similar ideas will be merged. Findings by major themes will be summarized. Important respondent narratives will be marked for future citations. A directory of the snakes found in the study blocks along with their local names will be created. Qualitative data analysis will be carried out using *NVivo* software (QSR International, UK).

## Discussion

SBE remains a major public health hazard predominantly in tropical countries [3]. In Asia up to two million people are bitten by snakes every year; the majority of them are women, children, and poor rural communities [32]. SBE has highly effective treatment and severe complications can be prevented with a correct dose of anti-venom [33]. Snakebite treatment remains a complex issue in countries like India due to various factors—poor community knowledge, ill-equipped hospitals, treatment accessibility, travel time, and reliance on faith healers for treatment [34]. Very little attention is drawn towards the assessment of basic knowledge of medical doctors on management of snakebite and its complications [35, 36].

A study conducted in India and Pakistan concluded that standard protocols and training are required to empower doctors to reduce snakebite mortality [35]. Another study from Bangladesh revealed that peripheral healthcare workers lack knowledge, experience, and adequate skills required to treat snakebite victims [37]. In a recent study from Bhutan, only 25% of health workers had adequate knowledge of snakebite management. However, 63% of medical doctors were found to have adequate knowledge compared to other healthcare workers [23]. A study in Haryana on farmers, teachers, medical residents, and students reported that only 13% participants, were aware of the 'big four' snakes [27]. Lack of appropriate knowledge on snakes, first aid, treatment, and prevention among the clinicians was also reported in northern Nigeria [38]. In Hong Kong, only 29% of clinicians were confident about treating snake bites [36].

India is a major manufacturer of anti-venoms. However, there is a paucity of satisfactory data regarding the number of anti-venom vials needed to reverse the clinical effects of SBE [39]. Passivity, inaction and supply-demand incoordination have resulted in both unavailability and inadequacy of anti-venom marginalizing the effective cure to SBE [40]. A study in Africa reported that compared to requirements, the number of effective treatments available for SBE might be as little as 2.5% [41]. In many African and Asian countries, anti-venom production is either at standstill or stopped altogether [42].

From the community point of view, preventing life-threatening snakebite incidences is the holy grail. Simple measures including use of bed-net while sleeping, use of torch and stick while walking in the dark, ban on open defecation practices, using knee length footwear in farms can aid in prevention [43, 44]. Community engagement using an integrated approach that involves strengthening the knowledge and awareness supplemented by efforts to tackle the sociocultural and economic barriers to seeking early healthcare can improve the management experience of many victims [6, 33]. Establishment of useful referral mechanisms between various stakeholders including community members, key influential persons in the area, administrative bodies and the healthcare system can go a long way in dealing with the SBE burden.

The National Snakebite Project will use a multi-sectoral, multi-stakeholder approach to reduce the burden of SBE. Encouraging community participation, healthcare provider empowerment, and wide-scale IEC activities are the strengths of the study. Health facility based retrospective and prospective data collection might result in underreporting of the SBE burden. However, an effort would be made to collect data from municipal corporation records, gram panchayat, crematorium or any other institutions in the study areas to ensure that maximum snakebite deaths are captured in addition to hospital records. Further, the evidence generated from the study may be useful to the Government of India for developing national strategies to reduce SBE burden. Model clinical snakebite management centers can be established all over the country based on the outcome of this study. The engagement of community members, local authorities, healthcare providers, herpetologists, social scientists, and public health experts is envisioned to accentuate the efforts for the prevention and control of snakebites in India.

## Supporting information

**S1 File. SPIRIT statement.**
(PDF)

**S2 File. Participant information sheet.**
(PDF)

**S3 File. Informed consent form.**
(PDF)

**S4 File. WHO trial registration data set.**
(PDF)

**S5 File. INSP ethics protocol.**
(PDF)

## Acknowledgments

The authors thank Dr. Geetanjali Sachdeva, Director, ICMR—NIRRCH, Mumbai, Dr Ashoo Grover, ICMR, New Delhi. The authors thank the members of the Technical Advisory Committee: Dr. Jaideep Menon, Dr. Yogesh Kalkonde, Dr. Dayal Majumdar, Dr. Milind Vyawahare, Dr. Kedar Bhide, and Dr. Joy Kumar Chakma for their critical inputs in the revision of data collection tools. INSP Project staff Ms. Bijaylaxmi Mohanty, Mr. Ganesh Bhad, Mr. Milind Gavhande and Mr. Jagdish Prasad Dash are acknowledged. The authors also thank the Public Health Departments in Maharashtra and Odisha states for providing administrative permissions for the study.

## Author Contributions

**Conceptualization:** Rahul K. Gajbhiye.

**Funding acquisition:** Rahul K. Gajbhiye.

**Investigation:** Rahul K. Gajbhiye, Itta Krishna Chaaithanya, Hrishikesh Munshi.

**Project administration:** Rahul K. Gajbhiye, Amarendra Mahapatra, Subrata Kumar Palo, Sanghamitra Pati, Arun Yadav, Manohar Bansode, Shashikant Shambharkar, Kanna Madavi, Himmatrao S. Bawaskar, Smita D. Mahale.

**Resources:** Rahul K. Gajbhiye.

**Supervision:** Rahul K. Gajbhiye.

**Writing – original draft:** Rahul K. Gajbhiye, Itta Krishna Chaaithanya, Hrishikesh Munshi, Ranjan Kumar Prusty.

**Writing – review & editing:** Rahul K. Gajbhiye, Itta Krishna Chaaithanya, Hrishikesh Munshi, Ranjan Kumar Prusty, Amarendra Mahapatra, Subrata Kumar Palo, Sanghamitra Pati, Arun Yadav, Manohar Bansode, Shashikant Shambharkar, Kanna Madavi, Himmatrao S. Bawaskar, Smita D. Mahale.

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
