## [Decision Letter · Decision Letter 0]

22 Aug 2022

PONE-D-22-12125

National snakebite project on capacity building of health system on prevention and management of snakebite envenoming including its complications in selected districts of Maharashtra and Odisha in India: a study protocol

PLOS ONE

Dear Dr. Gajbhiye,

Thank you for submitting your manuscript to PLOS ONE. After careful consideration, we have decided that your manuscript does not meet our criteria for publication and must therefore be rejected.

Specifically:

I am sorry that we cannot be more positive on this occasion, but hope that you appreciate the reasons for this decision.

Kind regards,

Narasimha Murthy Bhamidipati, Ph.D

Academic Editor

PLOS ONE

Additional Editor Comments:

(i) In the background of the study protocol, Maharashtra was not shown in the list of states contributing to snake-bite deaths. It is not clear why the authors chose to include Maharashtra in the study.

(ii)There is no justification of choosing a sample size of 160 MOs and 600 peripheral workers to be included in to the study.

(iii) It is not clear how the population will be trained: periodicity and tenacity

(iv) Retrospective study will underestimate the both morbidity and mortality. Prospective study is not well described to estimate the real incidence of both morbidity and mortality. The sample size needed to estimate the above parameters is not given nor described.

(v) The authors did not mention the type of statistical analysis or tools to be adopted in the study.

(vi) It is not clear what type of measured to be employed for snake bite prevention and thereby mortality.

(vii) There is no scientific gain of the study and therefore rejected.

Reviewers' comments:

Reviewer's Responses to Questions

**Comments to the Author**

1. Does the manuscript provide a valid rationale for the proposed study, with clearly identified and justified research questions?

Reviewer #1: Yes

2. Is the protocol technically sound and planned in a manner that will lead to a meaningful outcome and allow testing the stated hypotheses?

Reviewer #1: Partly

3. Is the methodology feasible and described in sufficient detail to allow the work to be replicable?

Reviewer #1: No

4. Have the authors described where all data underlying the findings will be made available when the study is complete?

Reviewer #1: No

5. Is the manuscript presented in an intelligible fashion and written in standard English?

Reviewer #1: No

6. Review Comments to the Author

You may also provide optional suggestions and comments to authors that they might find helpful in planning their study.

Reviewer #1: Comments:

I have two overarching comments regarding this protocol based on the PLOS ONE publication criteria (https://journals.plos.org/plosone/s/criteria-for-publication)

First, I don't believe criterion 3 ("Experiments, statistics, and other analyses are performed to a high technical standard and are described in sufficient detail.") has been met in this protocol. For instance, it's unclear how validity will be assessed in lines 209-231. There's also few details on how stratified analyses will be performed and how differences will be assessed (lines 306-309) and how changes between retrospective and prospective data (lines 295-296) and baseline to endline data (lines 314-315) will be evaluated. Ideally, these descriptions will propose methods and say which outcome measures will be used, e.g., means and standard deviations, proportions, odds ratios, etc., along with the statistical tests that will be used. I recommend providing greater detail on how the qualitative analyses will be performed (lines 319-320) or point to prior places in the protocol where those analyses are described.

Second, I am wavering on criterion 5 ("The article is presented in an intelligible fashion and is written in standard English."). While I think the article is intelligible, there are some grammatical errors throughout. For instance, I took the study setting paragraph and read it over:

1. (line 158) "more" should be "greater"

2. (line 159) "an" should be in front of "earlier"

3. (line 160) A greater problem compared to what? The rest of India?

4. (lines 161-2) I'd probably say "proposed" instead of "present" or maybe reword to say, "This multi-center study will be conducted…"

5. (lines 162-6) These two sentences are awkward and I suggest rewriting.

You should note that PLOS does not copy edit so I strongly suggest a thorough read of this manuscript before submitting again. If this manuscript has already been seen by an editing service, I strongly suggest finding a different one.

7. PLOS authors have the option to publish the peer review history of their article (what does this mean?). If published, this will include your full peer review and any attached files.

Reviewer #1: No

- - - - -

---

## [Author Response · Author response to Decision Letter 0]

27 Sep 2022

Point-to-point response to reviewer and editor comments

PONE-D-22-12125

National snakebite project on capacity building of health system on prevention and management of snakebite envenoming including its complications in selected districts of Maharashtra and Odisha in India: a study protocol

Reviewers' comments:

Sr. No. Comment Response Page No.

1 

‘I have two overarching comments regarding this protocol based on the PLOS ONE publication criteria (https://journals.plos.org/plosone/s/criteria-for-publication)’

(i) First, I don't believe criterion 3 ("Experiments, statistics, and other analyses are performed to a high technical standard and are described in sufficient detail.") has been met in this protocol. For instance, it's unclear how validity will be assessed in lines 209-231.

(ii) There's also few details on how stratified analyses will be performed and how differences will be assessed (lines 306-309) and how changes between retrospective and prospective data (lines 295-296) and baseline to end line data (lines 314-315) will be evaluated. Ideally, these descriptions will propose methods and say which outcome measures will be used, e.g., means and standard deviations, proportions, odds ratios, etc., along with the statistical tests that will be used. 

(iii) I recommend providing greater detail on how the qualitative analyses will be performed (lines 319-320) or point to prior places in the protocol where those analyses are described. 

The experiments, statistics, and other analyses are described in detail in the revised manuscript. 

The details of the methods of validating the research tools are described in the revised manuscript. 

The differences across strata will be investigated using appropriate statistical tests based on the nature of the variables. The Chi-square and Fisher’s Test would be used to compare categorical variables while the independent t-test would be used to compare continuous variables. To check the factors associated with the incidence of snakebites, bivariate regression models will be used and results will be presented through odds ratios. 

The secondary outcomes will include a reduction in the mean bite to needle time, a reduction in the proportion of snakebite victims referred to higher facilities for management, increase in the proportion of Medical Officers providing treatment as per Standard Treatment Guidelines (STG 2017) of Government of India, mean increase in the number of vials of anti-venom administered to victims, reduction in the proportion of cases with severe complications including rate of renal failure, amputations etc. 

A summative index will be developed from the knowledge test of health workers. Two sample t-test will be used to compare the change in the knowledge of Medical Officers and the healthcare workers after the training.

As suggested, we have added the details on qualitative data analysis in revised manuscript. 

Audios of the participants’ discussion will be recorded and notes will be taken. Transcripts will be proofread and then translated to English by the project staff. The transcriptions and translations will be cross-checked by senior project staff to ensure accuracy. A sample of the corrected translations will be read by the investigators to identify themes and subthemes using an inductive approach. Any additional themes or sub-themes after descriptive content analysis will also be identified. Disparities in the sub-themes or themes will be discussed by the investigators and consensus will be sought on the definitions of codes. Coding for the transcripts, based on the consensus definitions, will be carried out by two experienced staff members. All the coded transcripts will be reviewed for quality, consistency and accuracy. Broad themes based on similar ideas will be merged. Findings by major themes will be summarized. Important respondent narratives will be marked for future citations. A directory of the snakes found in the study blocks along with their local names will be created. 

Page 10-17

Page no. 

10-12

Line nos. 

198-243

Page 16

Line nos. 327-332

Page 16

Line nos. 335-340

Page 16

Line nos. 341-344

Page 16-17

Line nos. 348-359

2 Second, I am wavering on criterion 5 ("The article is presented in an intelligible fashion and is written in standard English."). While I think the article is intelligible, there are some grammatical errors throughout. For instance, I took the study setting paragraph and read it over:

1. (line 158) "more" should be "greater"

2. (line 159) "an" should be in front of "earlier"

3. (line 160) A greater problem compared to what? The rest of India?

4. (lines 161-2) I'd probably say "proposed" instead of "present" or maybe reword to say, "This multi-center study will be conducted…"

5. (lines 162-6) These two sentences are awkward and I suggest rewriting.

 We thank the reviewer for pointing out the grammatical errors in the manuscript.

The entire manuscript has been checked thoroughly and the errors have been corrected. 

Additional Editor Comments:

Sr. No. Comment Response Page No.

1 In the background of the study protocol, Maharashtra was not shown in the list of states contributing to snake-bite deaths. It is not clear why the authors chose to include Maharashtra in the study. We thank the editor for the comment. 

We have added the information in the background of study protocol (introduction section). Changes are made in revised manuscript. 

A nationally representative mortality survey conducted by Mohapatra B et al 2011, included Maharashtra in the high prevalence of snakebite envenomation (SBE) group with an age standardized mortality rate of 3.0 per 100000 people. Except the high prevalence states, for the rest of the country the age standardized mortality rate was 1.8 per 100000 people during the same period. In the same study, Maharashtra was also reported to have the fifth largest number of deaths due to snakebite among the high prevalence states. 

A study by Suraweera W et al 2020, estimated 56000 SBE deaths in Maharashtra from 2001-14. However, only a 10% coverage of the actual snakebite burden being captured in the government data in Maharashtra was also reported indicating gross underestimation of morbidity and mortality in Maharashtra.

In a prior study conducted by the authors in the tribal region of Dahanu, district Palghar, Maharashtra, India, the annual incidence of snakebite was 36 per 100,000 populations (January to December 2014), with a case fatality rate of 4.5%. However, this study used retrospective case information gathered from one subdistrict hospital in the Palghar district's tribal block (Gajbhiye R et al 2019). It is also clear that rural and tribal populations had insufficient and varying understanding and perceptions of snakebites (Chaaithanya IK 2020).

The government of India response to a question raised in Parliament on 07th February 2020, reported data on snakebite from 2016 to 2018. Maharashtra reported 65044 cases (third highest in the country after West Bengal and Andhra Pradesh) with 134 deaths. A copy of the reply of Ministry of Health and Family Welfare, Government of India is attached. 

Dr. Himmatrao Saluba Bawaskar (HSB), Co-Principal Investigator of the study and co-author in the manuscript is a global authority on snakebite research. HSB has been working on snakebite envenomation for the last four decades. HSB has published several articles in world’s leading journals highlighting the high burden of snakebite cases in Maharashtra State. 

The lead author and Principal Investigator Rahul Gajbhiye (RG) and HSB are working closely with the Government of Maharashtra for more than 10 years. Public Health Officials from Govt. of Maharashtra are involved at all steps of the execution of this study. Based on the outcomes of the study, the model for reducing snakebite mortality and morbidity could be replicated in other affected districts by the Government.

A Nationwide Study to estimate incidence, mortality, morbidity, and economic burden due to snakebites in India is ongoing in 13 states, where Maharashtra was identified as one of the states. The protocol for this study is recently published in PLOS One.

We hope that the abovementioned facts and evidence will convince the editor why Maharashtra was included in the study. 

Page 5-6

Line nos. 87-93

Page 5

Line nos. 81-83

2 There is no justification of choosing a sample size of 160 MOs and 600 peripheral workers to be included in to the study. All the Medical Officers and Peripheral Health Workers from the study blocks who agree to participate will be included in the study. The total number of MOs and peripheral health care workers working in study blocks are 160 and 600 respectively. This information was collected from the state health departments of Maharashtra and Odisha states. Page 14

Line nos. 280-284

3 It is not clear how the population will be trained: periodicity and tenacity After the pre-intervention Focus group discussions in the community, an intervention phase with the IEC campaign and community discussions/talks would be adopted. Based on the gaps found in the baseline survey (FGDs), educational interventions in the form of printed teaching materials will be prepared and shared with the community.

FGDs will be conducted in the study areas to evaluate the post-intervention community knowledge with the same methodology as the pre-intervention phase. Page 13-14

Line nos. 260-277

Page 15

Line nos. 308-309

4 Retrospective study will underestimate the both morbidity and mortality. Prospective study is not well described to estimate the real incidence of both morbidity and mortality. The sample size needed to estimate the above parameters is not given nor described. We will be conducting both prospective and retrospective data collection as mentioned in the manuscript. We will cover all the health facilities in the study blocks to cover incidence and case fatality in last two years to measure baseline data and to estimate the incidence and case-fatality. We appreciate the concern of underestimation of mortality and morbidity. To overcome this challenge and to account for the missing data, additional data will be collected from the following institutions - tertiary care hospitals and medical colleges, municipal corporations, gram panchayat, block development officers, registrar of births and deaths, ASHA workers. This strategy will be adopted for both retrospective and prospective data collections. 

Since we are covering all cases from all health facilities there is no question of sample size estimation Page 12

Line nos. 247-249

Page 15

Line nos. 311-314 

5 The authors did not mention the type of statistical analysis or tools to be adopted in the study We have added the details of statistical analysis in the revised manuscript. Page 16-17

Line nos. 325-360

6 It is not clear what type of measured to be employed for snake bite prevention and thereby mortality The prevention measures to be employed for snakebite prevention will be as per the Standard Treatment Guidelines, (STG,2017). 

Our earlier observations demonstrate that snakebite incidents in the community are affected by a variety of factors that are location and population specific. Understanding community awareness, perception, and first aid practices of snakebites are critical for developing the most effective preventative interventions. In addition to this, capacity building of local medical officers is also important to prevent snakebite mortality. The study is proposed to address these issues, thereby appropriate preventive measures would be recommended based on the information gathered from the baseline survey (FGD) in addition to prevention methods recommended in STG, 2017. Page 13-14

Line nos. 275-277

7 There is no scientific gain of the study and therefore rejected The proposed project will provide the following scientific benefits. 

1. This is the first large-scale study to generate evidence on the implementation of Standard Treatment Guidelines, 2017. Based on the evidence generated from this study, a policy recommendation can be provided to the Central and State governments on the implementation of STG, 2017 in most affected areas.

2. The evidence generated from the study will be useful for the finalization of the National Snakebite Management Protocol taking into consideration any regional factors emerging from the study. 

3. The study will generate evidence on community empowerment and capacity building of the Health System for improved management of snakebites. 

4. The study will provide the information on incidence, and case fatality rate of snakebites in the selected study sites in 2 high burden states in India. 

5. The study will provide information on the knowledge, health-seeking practices, traditional practices, and myths of the community on snakebites in selected study sites. 

6. The study will generate evidence on the availability of ASV, utilization of ASV, and evidence on adverse reactions to ASV. 

7. The study will generate evidence on knowledge of Medical Officers and peripheral health care workers on snakebite in selected study sites in Maharashtra and Odisha. Based on the evidence generated from the study, appropriate interventions may be suggested to the state health departments on capacity building. 

8. The study will generate evidence on the relationship between the community and the local health system to understand the local knowledge and perception of Snakebite. 

9. Based on the study findings, national-level solutions to reduce snakebite mortality would be devised.

10. The study will establish Model clinics for the treatment of snakebites in study blocks.

---

## [Decision Letter · Decision Letter 1]

23 Dec 2022

PONE-D-22-12125R1

National snakebite project on capacity building of health system on prevention and management of snakebite envenoming including its complications in selected districts of Maharashtra and Odisha in India: a study protocol

PLOS ONE

Dear Dr. Gajbhiye,

Thank you for submitting your manuscript to PLOS ONE. After careful consideration, we feel that it has merit but does not fully meet PLOS ONE’s publication criteria as it currently stands. Therefore, we invite you to submit a revised version of the manuscript that addresses the points raised during the review process.

Reviewers have commented favorably on the manuscript but nonetheless reviewer #2 has raised major concerns and recommended significant revisions.

We look forward to receiving your revised manuscript.

Kind regards,

Karen de Morais-Zani

Academic Editor

PLOS ONE

Journal Requirements:

“The authors also thank the Public Health Departments in Maharashtra and Odisha states for providing administrative permissions and support for the study. Dr. Rahul K Gajbhiye is an awardee of the DBT-Wellcome India alliance clinical and public health intermediate fellowship (Grant no. IA/CPHI/18/1/503933).”

“This project is funded by the Indian Council of Medical Research (ICMR), (no: 58/6/NTF-Snakebite/2019-NCD-II). The funding agency has no role in study design, collection, management, analysis and interpretation of data; writing of report; and the decision to submit the report for publication. The funding agency will have no authority over any of these activities.”

“This project is funded by the Indian Council of Medical Research (ICMR), (no: 58/6/NTF-Snakebite/2019-NCD-II). The funding agency has no role in study design, collection, management, analysis and interpretation of data; writing of report; and the decision to submit the report for publication. The funding agency will have no authority over any of these activities”

Additional Editor Comments (if provided):

Reviewers' comments:

Reviewer's Responses to Questions

**Comments to the Author**

1. Does the manuscript provide a valid rationale for the proposed study, with clearly identified and justified research questions?

Reviewer #1: Yes

Reviewer #2: Partly

2. Is the protocol technically sound and planned in a manner that will lead to a meaningful outcome and allow testing the stated hypotheses?

Reviewer #1: Yes

Reviewer #2: Partly

3. Is the methodology feasible and described in sufficient detail to allow the work to be replicable?

Reviewer #1: Yes

Reviewer #2: No

4. Have the authors described where all data underlying the findings will be made available when the study is complete?

Reviewer #1: Yes

Reviewer #2: Yes

5. Is the manuscript presented in an intelligible fashion and written in standard English?

Reviewer #1: Yes

Reviewer #2: Yes

6. Review Comments to the Author

You may also provide optional suggestions and comments to authors that they might find helpful in planning their study.

Reviewer #1: I thank the authors for carefully considering my comments on the prior draft. The expanded statistical methods section is noted and I believe is sufficient for a study protocol. Statistical methods sections are hard to write in protocols because (1) it's hard to predict what will be needed for analyses and (2) detail is needed to understand the proposal but too much detail may result in protocol revisions.

Regarding criterion 5, I see the changes to the study setting paragraph, but based on the tracked changes version of the manuscript I'm not convinced that the authors reviewed the whole manuscript in detail. I'd expect to see small changes throughout the manuscript. That said, maybe these were not recorded and, as I said prior, the article is intelligible, which is most important.

Best wishes and good luck with your study.

Reviewer #2: The authors listed underestimated reports and deaths in consequence of snakebite envenoming, especially in Maharashtra state, India. However, to build a national snakebite project for strengthening health system on prevention in management, one important component is a reliable epidemiological surveillance system. In this matter, no mention was given to snakebite envenoming compulsory notification and strategies to collect robust surveillance data

Considering, as authors referred, that only 10% coverage of the actual snakebite burden have being captured by the official data (ref.10], it would be a crucial element of the project to increase the capillarity of the system to collect snakebite envenoming cases.

The morbimortality data, as well as risk factors analysis of deaths and disabilities should be the basis to guide program interventions and sharpen public communications. Thus, which strategies would be designed to generate sufficient quality data that matches the objectives of the protocol?

Phase I was described as been composed of retrospective data collection from public health facilities, and focal groups discussions in the community. It is not clear the selection criteria for the health facilities. Would be at random? How representativeness will be guaranteed to assure that different categories of health facilities would be represented.

Still in Phase I, how focal groups will be selected in the community to participate in the study? Community leaders? Family members of a snakebite patient? Neighbors? Extended family?

Pre and post-training evaluation will be done to assess knowledge but it is not clear how long before and after the distribution of educational materials distributed in key places. Another question is how participants will be selected; will it be at random?

Methodology of the intervention does not supply sufficient information to understand how deep will the authors explore this issue.

Validation of the educational tools seems not to be included.

7. PLOS authors have the option to publish the peer review history of their article (what does this mean?). If published, this will include your full peer review and any attached files.

Reviewer #1: No

Reviewer #2: No

---

## [Author Response · Author response to Decision Letter 1]

13 Jan 2023

RESPONSE TO REVIEWER’S COMMENTS 

PONE-D-22-12125R1

Title: National snakebite project on capacity building of health system on prevention and management of snakebite envenoming including its complications in selected districts of Maharashtra and Odisha in India: a study protocol

Journal requirement 

Response: The manuscript has been revised to meet PLOS ONE’s style requirements as per the templates provided in the comment.

Paragraph formatting and font size change are done throughout the manuscript.

Title and author page revisions on page no. 1-2

“The authors also thank the Public Health Departments in Maharashtra and Odisha states for providing administrative permissions and support for the study. Dr. Rahul K Gajbhiye is an awardee of the DBT-Wellcome India alliance clinical and public health intermediate fellowship (Grant no. IA/CPHI/18/1/503933).”

“This project is funded by the Indian Council of Medical Research (ICMR), (no: 58/6/NTF-Snakebite/2019-NCD-II). The funding agency has no role in study design, collection, management, analysis and interpretation of data; writing of report; and the decision to submit the report for publication. The funding agency will have no authority over any of these activities.”

Response: Funding-related text has been removed from the manuscript (Page no. 20-21, Acknowledgement section). 

The amended funding statement has been included in the cover letter.

“This project is funded by the Indian Council of Medical Research (ICMR), (no: 58/6/NTF-Snakebite/2019-NCD-II). The funding agency has no role in study design, collection, management, analysis and interpretation of data; writing of report; and the decision to submit the report for publication. The funding agency will have no authority over any of these activities”

Response: The amended funding statement has been included in the cover letter. Manuscript Page no. 21, Line no. 454-457

Response: The amended data availability statement has been included in the cover letter. Manuscript Page no. 23, Line no. 501-504

Response: The ethics statement has been moved to the Methods section of the manuscript as per the comment.

Page No. 15-16, Line No. 329-341

Reviewer #1: 

6. I thank the authors for carefully considering my comments on the prior draft. The expanded statistical methods section is noted and I believe is sufficient for a study protocol. Statistical methods sections are hard to write in protocols because (1) it's hard to predict what will be needed for analyses and (2) detail is needed to understand the proposal but too much detail may result in protocol revisions.

Response: We thank the reviewer for critically reviewing the statistical analysis section and providing constrictive feedback to improve it. 

7. Regarding criterion 5, I see the changes to the study setting paragraph, but based on the tracked changes version of the manuscript I'm not convinced that the authors reviewed the whole manuscript in detail. I'd expect to see small changes throughout the manuscript. That said, maybe these were not recorded and, as I said prior, the article is intelligible, which is most important.

Response: The manuscript was thoroughly revised to make it intelligible and small typos and writing errors were corrected after reviewer’s feedback. 

Reviewer #2: 

8. The authors listed underestimated reports and deaths in consequence of snakebite envenoming, especially in Maharashtra state, India. However, to build a national snakebite project for strengthening health system on prevention in management, one important component is a reliable epidemiological surveillance system. In this matter, no mention was given to snakebite envenoming compulsory notification and strategies to collect robust surveillance data. Considering, as authors referred, that only 10% coverage of the actual snakebite burden have being captured by the official data (ref.10], it would be a crucial element of the project to increase the capillarity of the system to collect snakebite envenoming cases. The morbimortality data, as well as risk factors analysis of deaths and disabilities should be the basis to guide program interventions and sharpen public communications. Thus, which strategies would be designed to generate sufficient quality data that matches the objectives of the protocol?

Response: We thank the reviewer for raising the concern. The proposed study focuses on capacity building of health systems on prevention and management of snakebite envenomation and would be useful for updating the national protocol for snakebite treatment in India and providing regional inputs for the same. 

Regarding the objectives of the present study, strategies to generate quality data, including validation of research tools, review and approval from Technical Advisory Committee, training of project staff before study implementation, IEC material dissemination, focus group discussions, health facility assessments are provided in sufficient detail in the protocol. 

We agree with the reviewer about the pressing need to have a robust surveillance system to capture maximum cases and deaths due to snakebite. However, that remains beyond the scope of the present study. Community empowerment using culturally appropriate Information, Education and Communication (IEC) material is an integral component of the present study. Improving knowledge of the communities regarding the prevention and first aid of snakebite along with advocating healthcare seeking from trained medical practitioners is indirectly expected to improve their healthcare seeking behavior and contribute to improving the number of cases and deaths being reported at the health facilities in the study sites. Through training programs, the study also aims to empower the medical doctors and healthcare workers for the better prevention, diagnosis and management of snakebite envenomation. Availability of trained staff is also expected to improve the confidence that people have in the health systems and motivate them to seek appropriate care. 

The ICMR task force project by Menon et al (doi.org/10.1371/journal.pone.0270735), have proposed a community-level surveillance for snakebites covering 31 districts in 13 states of India including Maharashtra to obtain the annual incidence of snakebites from the community. This study would help in building up a strong surveillance system in India and provide morbimortality data of snakebite envenomation. A significant difference, if any, in the findings of the Menon et al., study and the health system data can further pave way for recommending compulsory notification of snakebite envenomation in India, as suggested by the reviewer.

9. Phase I was described as been composed of retrospective data collection from public health facilities, and focal groups discussions in the community. It is not clear the selection criteria for the health facilities. Would be at random? How representativeness will be guaranteed to assure that different categories of health facilities would be represented.

Still in Phase I, how focal groups will be selected in the community to participate in the study? Community leaders? Family members of a snakebite patient? Neighbors? Extended family?

Response: In the proposed study, we are going to include all the public health facilities in the study areas where snakebite management is expected to be provided. This includes all the Primary Health Centres, Rural Hospitals / Community Health Centres, Sub District Hospitals, District Hospitals. Medical Colleges where critical cases are referred from the study sites will also be included to ensure quality retrospective data collection. This has been included in the revised manuscript (Page no. 12-13, Line no. 250-255). 

Additional data on snakebite deaths will also be collected from local authorities, panchayat office, and registrar of births and deaths so that the maximum number of deaths are captured. This section is already present in the manuscript (Page no. 13, Line no. 255-256). 

Regarding selection of focal groups in the community, the selection process has been revised in the manuscript (Page no. 13, Line no. 264-269)

10. Pre and post-training evaluation will be done to assess knowledge but it is not clear how long before and after the distribution of educational materials distributed in key places. Another question is how participants will be selected; will it be at random?

Methodology of the intervention does not supply sufficient information to understand how deep will the authors explore this issue.

Validation of the educational tools seems not to be included.

Response: The study proposes to do a post-intervention evaluation after two months (Page 15, Line no. 320-322). The methodology including the selection of participants for the post-intervention evaluation will remain the same as that of pre-intervention assessment. The educational intervention would be implemented in the study sites by trained and experienced research staff. All the interventional material will be reviewed and endorsed by the technical advisory committee of the study. This has been added in the revised manuscript (Page no. 14, Line no. 280-282). 

Mock or trial intervention procedures will be implemented to know the community's understanding capacity accordingly further intervention procedures will be carried out. This has been added in the revised manuscript (Page no. 13-14, Line no. 278-280). 

Education tools will be validated with the help of the Technical advisory committee and local herpetologists and subject experts. This has been added in the revised manuscript (Page no. 14, Line no. 280-282). 

We thank the editoral board, editor and reviewes for giving an opportunity to revise the manuscript.

---

## [Editor Report · Decision Letter 2]

2 Feb 2023

National snakebite project on capacity building of health system on prevention and management of snakebite envenoming including its complications in selected districts of Maharashtra and Odisha in India: a study protocol

PONE-D-22-12125R2

Dear Dr. Gajbhiye,

We’re pleased to inform you that your manuscript has been judged scientifically suitable for publication and will be formally accepted for publication once it meets all outstanding technical requirements.

Kind regards,

Karen de Morais-Zani

Academic Editor

PLOS ONE
---

## [Editor Report · Acceptance letter]

9 Feb 2023

PONE-D-22-12125R2 

National snakebite project on capacity building of health system on prevention and management of snakebite envenoming including its complications in selected districts of Maharashtra and Odisha in India: a study protocol 

Dear Dr. Gajbhiye:

I'm pleased to inform you that your manuscript has been deemed suitable for publication in PLOS ONE. Congratulations! Your manuscript is now with our production department. 

Kind regards, 

on behalf of

Dr. Karen de Morais-Zani 

Academic Editor

PLOS ONE